# Parental Burnout in the Context of Special Needs, Adoption, and Single Parenthood

**DOI:** 10.3390/children10071131

**Published:** 2023-06-29

**Authors:** Isabelle Roskam, Moïra Mikolajczak

**Affiliations:** Department of Psychology, UCLouvain, Place Cardinal Mercier 10, B-1348 Louvain-la-Neuve, Belgium; moira.mikolajczak@uclouvain.be

**Keywords:** exhaustion, disability, illness, disorder, adopted child, adoptive parent, single mom, single dad

## Abstract

Although early work on parental burnout in the 1980s did not link parental burnout to specific parenting situations, the idea that parents affected by specific vulnerability factors were particularly at risk of burnout quickly emerged. Based on the etiological model of parental burnout (i.e., the balance between risks and resources), the objective of this study was to investigate if there were specific vulnerability factors that significantly increase the risk of parental burnout. 2563 parents participated in the study. We compared parents with a child with special needs (*n* = 25), parents who had adopted a child (*n* = 130), and single parents (*n* = 109), to strictly matched control groups of parents. Parents with a child with special needs displayed higher burnout and lower balance between risk and resources than the control group; parents with an adopted child had similar levels of both parental burnout and balance; and single parents had higher parental burnout but similar balance. Parents who combined specific several vulnerability factors were at greater risk of burnout. Single parenthood and adoption do not in themselves trigger a process leading de facto to other risk factors, but having a child with special needs does.

## 1. Introduction

It has previously been shown that parents who are chronically stressed in their parenting situation are more vulnerable to parental burnout, a disorder characterized by physical and/or emotional exhaustion as a parent, emotional distancing from children, feelings of being fed up in the parental role, and a sense of contrast with the previous parental self [1,2]. To what extent, if any, does having a child with special needs, having adopted a child, or being a single parent increase the risk of parental burnout?

Early work on parental burnout in the 1980s did not link parental burnout to specific parenting situations. For instance, Lanstrom [3] wrote a book on Christian parents who suffered physical and emotional exhaustion, guilt, and feelings of inadequacy in dealing with the problems of raising healthy children. The book by Procaccini and Kiefaver [4] addressed how parents, in general, coped with the problems, frustration, and mental, physical, and emotional exhaustion of raising a child. Pelsma [5] conducted the very first empirical study on parental burnout among nonworking mothers.

However, the idea that parents affected by a specific vulnerability factor were particularly at risk of burnout quickly surfaced. The second, third and all subsequent studies conducted during the next 20 years focused on parents of children with special needs, in particular autism and mental retardation [6], suffering from serious illnesses such as Type 1 diabetes mellitus, or recovering from a brain tumor [7,8,9,10,11]. These studies all showed that such vulnerability factors significantly increased the risk of parental burnout. More recent studies have supported these findings (e.g., [12,13,14]).

Considering that having adopted a child or single parenthood are vulnerability factors to parental burnout, is also consistent with studies showing that parental stress and depressive symptoms are higher in single mothers than in partnered mothers (e.g., [15]), and that they have a poorer physical and mental health (e.g., [16]). It is also a widespread assumption in the media and on blogs (e.g., [17,18,19]). A similar vulnerability assumption was assumed for adoptive parents [20].

Are things really that straightforward? Recent studies which reported the effect of having a child with special needs and of single parenthood on parental burnout, have tempered the above view. Regarding parents of children with special needs, Gérain and Zech [21] showed that only having a child with several special needs or having at least two children with special needs would significantly increase the risk of parental burnout. Regarding adoptive parents, Piraux and Mehauden [22] showed that these parents were only slightly more at risk of parental burnout than control parents. Finally, regarding single parenthood, several studies showed that its effect on parental burnout was weak [23,24], even in high-stress situations such as during the COVID-19 pandemic [25], or even protective [26]. To date, no study has investigated whether or not having an adopted child increases the risk of parental burnout.

What can we conclude then? Deciding on the basis of previous studies is difficult, mainly because most studies rely on small samples, do not control for possible confounding variables, and do not take into account the combination of different vulnerabilities in some families. Furthermore, in no previous study has the control group been demographically matched. To address the question of whether there are specific vulnerability factors for parental burnout in a convincing manner, it is necessary both to have a theoretical framework and to use a methodology that takes into account the confounding variables and the combination of vulnerabilities in certain families. This was the objective of the present research.

### 1.1. The Etiological Framework of Parental Burnout

The etiological model of parental burnout posits that parental burnout results from a chronic imbalance between the risks the parent faces and the resources available to him/her [2]. In this balance between risks and resources (BR^2^), any risk can be compensated for by any resource(s) as long as the sum of their weight is equivalent to the weight of the risk(s) in the balance. Research has shown, for example, that parenting perfectionism can be compensated for by emotional competencies [27,28], and that an objectively demanding situation for parents, such as the lockdown of parents with their children during the COVID-19 pandemic, can be compensated for by efficient emotion regulation abilities [29].

According to the BR^2^ etiological theory, we can hypothesize that a vulnerability factor such as having a child with special needs, having adopted a child, or being a single parent, can be compensated for either by a resource of equivalent weight, or by a combination of smaller resources. For example, having a child with special needs or having an adopted child could be compensated for by increased financial, instrumental, informational, and emotional support from specialized services and parent associations, and/or by increased informal support from relatives. In the same way, single parenthood could also be compensated for by social support together with encouraging children to be more autonomous from an early age.

The foregoing would explain why parents with a specific vulnerability factor do not systematically have a higher level of burnout than other parents, and why a vulnerability factor alone explains only a limited part of the variance in parental burnout [23,24,25]. The presence of several vulnerability factors will, however, be more difficult to compensate for with resources because of their cumulative effect [30,31]. This would explain the finding of Gérain and Zech [21] that the combination of several vulnerability factors (i.e., having at least two children with special needs), significantly increases the risk of parental burnout.

The cumulative effect of two different vulnerability factors has been considered in previous studies. For example, whereas Shireman [32] concluded that the cumulative effect between single motherhood and having adopted a child was a myth, Díez, González [33] reported an interactive effect of these two factors on children’s psychosocial adjustment and social competence. The stress of single mothers was also seen as the result of a certain number of stressors occurring in combination, namely having a child with special needs [30]. The same observation was made for adoptive parents facing specific challenges such as rearing children with special needs [20].

The BR^2^ etiological theory also leads to the prediction that, in the absence of any of the specific vulnerability factors mentioned above, a parent can present a high level of burnout if he or she combines other risks (e.g., having to reconcile parenting with complicated work schedules, having several young children, being in disagreement about childrearing values with the co-parent, or having low parental self-efficacy beliefs) without enough resources to compensate for them. In other words, the absence of a specific vulnerability factor such as having a child with special needs or single parenthood, does not necessarily prevent the risk of parental burnout. This would explain the prevalence of parental burnout found in the general population i.e., around 5%, and up to 8–10% in some Western countries [34].

### 1.2. Methodological Approach

Studying the true influence of any specific vulnerability factor is challenging because of the co-occurrence of multiple risk factors which can be interrelated. For example, having a child with special needs may be related to financial difficulties, and single parenthood may be associated with lower household income. The co-occurrence of multiple risk factors is responsible for the presence of confounding demographic effects that are not considered in most studies. To study the effect of a particular risk factor, it is necessary to take into account the confounding demographic variables, measure them, and control their effect in the model tested. An even better approach consists in comparing samples that are demographically matched for all confounding variables, to estimate the effect of a particular vulnerability factor, all other things being equal.

To investigate the true influence of vulnerability factors, the possible co-occurrence of multiple risk factors which are not usually interrelated must also be taken into account. For example, a parent with a child with a disability or adopted child may sometimes become a single parent following the death of his or her spouse or separation, or a parent may have adopted a child and also have a biological child (or even the adopted child) with special needs. The presence of vulnerability factors that are not necessarily co-occurring is rarely assessed and taken into account in studies. This requires the collection of data in several at-risk samples at the same time and considering the combination of these risks as an additional vulnerability factor, whose effect must also be estimated in isolation from the effect of each risk.

### 1.3. The Current Study

In this study, we assessed parental burnout and the balance between risks and resources, i.e., BR^2^, in three specific samples: parents with a child with special needs, parents who had adopted a child, and single parents. We then compared these parents to control groups of parents, taking care to (1) identify confounding variables, (2) strictly match the specific and the control groups for all confounding demographic variables, and (3) estimate the effect of the combined presence of a child with special needs and/or an adopted child and/or single parenthood, on both the balance between risks and resources and parental burnout.

Based on previous empirical research, it is currently difficult to predict whether the level of parental burnout will be significantly different between parents in the three specific samples, and control parents. However, based on the BR^2^ etiological model which not only captures the accumulation of stressors but also the accumulation of resources, we expect that

**H1.** 
*The level of burnout of the parents in the three specific samples will on average not differ from that of the parents in the control groups.*


**H2.** 
*Although we cannot rely on previous studies (e.g., [30,33]) that have estimated the accumulation of risk but not the accumulation of resources available to parents, the BR^2^ etiological model allows us to predict that the level of the balance between risks and resources of the parents from the three specific samples will on average not differ to that of the parents in the control groups.*


**H3.** 
*Finally, in line with both previous evidence and the BR^2^ etiological model, we expect that the combination of specific vulnerability factors (i.e., the combined presence of a child with special needs and/or an adopted child and/or single parenthood), will significantly increase the risk of parental burnout and the risk of imbalance between risks and resources.*


## 2. Materials and Methods

### 2.1. Participants

Data were collected from a sample of 2563 French- and Dutch-speaking parents (M_age_ = 43.33 years, SD_age_ = 7.98 years, range: 19–73, 84% mothers, 70% French-speaking) with at least one child still living at home. They had two children on average (range 1–7). Most of them were living in Belgium (74.5%), others in France (14%) or Switzerland (10%), and the rest in other French-speaking countries (<2%); 44.5% of them were living with a partner, 42.5% were single parents, 247 were in step-families, and 86 were in same-sex partnerships; 6.5% (*n* = 168) had one child with specific needs, and 1.5% (*n* = 44) had at least two children with specific needs. 9% (*n* = 230) had adopted a child. As regards the presence of two of the three vulnerability factors under study, 1.7% were single parents having an adopted child, 3.6% were single parents having a child with special needs, and 0.7% were parents having an adopted child and a child with special needs. As regards the presence of the three vulnerability factors under study, 44.3% of parents had no vulnerability, 50.5% had one, 4.5% had two, and 0.7% had three.

Thirty percent reported having completed up to 12 years of education from the first grade onwards, 32% had a university or college degree, 30% had a master’s degree, and 8% had a Ph.D. or MBA degree. With regard to their working status, 24% did not work, 32% had a part-time job, and 44% had a full-time job. Their net monthly household income was less than €2500 for 46% of the parents, between €2500 and €3999 for 29%, and higher than €4000 for 25%.

### 2.2. Procedure

Data were collected online using Qualtrics software with the forced choice option, ensuring a dataset with no missing data. The questionnaire was available in French and Dutch. Parents had to be at least 18 years old and have at least one child still living at home. Informed consent preceded the survey. Participation was voluntary; parents could withdraw from the study at any time without giving reasons; data were anonymous.

The study was presented as investigating factors of parental fulfillment and exhaustion. Parental burnout was not mentioned, in order to avoid participant self-selection bias. Participants were recruited between November 2019 and October 2021 via social networks, word of mouth, and with the help of the largest Belgian mutual health insurance company (insuring 42% of the Belgian population). 4040 parents participated in the study, and 2563 of them completed the sociodemographic, parental burnout, and balance between risk and resource factors measures. Note that the question of having a child with special needs was investigated in a subsample resulting in 1467 parents who completed the sociodemographic, parental burnout, and balance between risk and resource factors measures. Participants who completed the survey had the opportunity to participate in a lottery to win €200. To do so, they had to leave their email address in another questionnaire disconnected from the first one to ensure anonymity.

### 2.3. Measures

This study was part of a larger collaborative project aiming to study parental burnout in several specific populations (parents with a child with special needs, parents who had adopted a child, parents with a gifted child, single parents, same-sex parents, and low-SES parents). Other hypotheses have already been tested in same-sex parents [35], parents with gifted children [36], or parents living in poverty [37]. Only the instruments relevant to the present research questions are presented here.

Participants were first asked about the following: their sex (mother vs. father); their age; the number of children; the age of the children; the family type (two-parent family; single-parent family, step-family; two same-sex parents); their educational level (up to secondary level, university or college, master’s degree, Ph.D. or MBA degree); their working status (not in paid work, part-time job, full-time job); and their net monthly household income (<€2500, €2500–3999, €4000–5499, €5500–7000, >€7000).

Parental burnout was assessed with the Parental Burnout Assessment [PBA, 1], a 23-item questionnaire assessing the four core symptoms of parental burnout: emotional exhaustion (9 items) (e.g., I feel completely run down by my role as a parent), contrast with previous parental self (6 items) (e.g., I tell myself I’m no longer the parent I used to be), loss of pleasure in one’s parental role (5 items) (e.g., I don’t enjoy being with my children) and emotional distancing from one’s children (3 items) (e.g., I am no longer able to show my children that I love them) using a 7-point frequency scale (never, a few times a year, once a month or less, a few times a month, once a week, a few times a week, every day). In the current study, Cronbach’s alpha was 0.97.

The Balance Between Risks and Resources [BR^2^, 2] was assessed by means of 39 bipolar rating scales encompassing 11 levels, i.e., from −5 to +5 going through 0. The negative pole represented the risk while the positive pole represented the corresponding resource, for example, −5: “My children are so demanding of me that I don’t have a moment for myself”, or +5: “My children are demanding but I still have time free to do other things”. The total score was computed by summing the 39 items (min. −195, max. 195) so that positive scores indicated that the parent had more (or more significant) resources than risks, negative scores indicated that the parent had more (or more significant) risks than resources, and zero scores indicated that the parent had the same level of risks and resources. For single parents, the 9 items referring to the partner were removed from the survey. Their global score was obtained by summing the remaining 30 items (min. −150, max. 150) (which statistically amounts to nullifying the significance of removed items in the balance). The BR^2^ score was then standardized for all parents in order to have the same metric. Because the presence of one risk factor was not expected to be necessarily associated with the presence of another risk factor (e.g., a parent may have difficulties reconciling family and work without being a perfectionist parent), estimating an index of reliability such as Cronbach’s alpha is not relevant for BR^2^.

### 2.4. Statistical Analyses

Data analyses were performed with Stata 17 software [38]. We first conducted preliminary analyses to check for outliers and normality. We found no outlier. Skewness and kurtosis values were within the thresholds of |2.0| and |9.0| respectively, at which the results of parametric tests remain robust, as shown by Schmider and Ziegler [39]. Second, we tested for confounding effects in the three specific samples separately. We used t-tests for continuous variables and logistic regressions for categorical variables. We obtained Cohen’s d effect sizes for continuous variables and Cramer’s V for categorical ones. Then, based on the confounding demographic variables, we strictly matched the parents from the three specific samples with parents from the control groups.

The main analyses aimed to compare the level of parental burnout and the level of the balance between risks and resources, between parents from each of the three specific samples, and control groups of parents, using t-tests (i.e., H1 and H2). We also obtained Cohen’s d-effect sizes. We then correlated the balance between risks and resources and the level of parental burnout in the three specific samples and the control groups of parents. Finally, we tested the mean differences both in parental burnout and in the parents’ balance between risks and resources, according to the combination of specific vulnerability factors, with ANOVAs in the pooled sample (*n* = 2563). In order to obtain Cohen’s d effect sizes, we then compared the groups two by two (i.e., H3).

We interpreted Cohen’s d effect sizes according to the common thresholds of 0.2–0.49 for a small effect (corresponding to 58% of the parents from the specific sample being below/over the mean of the control group), 0.5–0.8 for a medium effect (corresponding to 69% of the parents from the specific sample being below/over the mean of the control group), >0.8 for a large effect (corresponding to 79% of the parents from the specific sample being below/over the mean of the control group). For Cramer’s V, values ranging from 0 to 0.3 must be considered as weak, from 0.4 to 0.5 as medium, and above 0.5 as strong.

Data and syntax are available on OSF https://osf.io/e4xw5/ (accessed on 17 August 2022).

## 3. Results

### 3.1. Preliminary Analyses

We identified six confounding demographic factors for the parents with a child with special needs: parent’s age, number of children, family type, parent’s educational level, working status, and income. In particular, parents of children with special needs were older, t(1465) = −5.17, *p* < 0.001, d = 0.27, had more children, t(1247) = 1.99, *p* = 0.023, d = 0.38, were less likely to be a single parent than to live with the other parent, b = −1.44, *p* < 0.001, V = 0.23, had a lower educational level, t(1465) = 2.85, *p* = 0.004, d = 0.21, were less likely to work full-time than to have no job, b = −0.49, *p* = 0.007, V = 0.08, and had a higher income, t(1465) = 4.92, *p* = 0.004, d = 0.37, than parents without a child with special needs. Considering the confounding effects of demographic factors, we successfully matched 25 pairs of parents. We checked the differences between matched and non-matched parents. Matched parents displayed a higher level of PB than non-matched ones, t(2561) = 2.55, *p* = 0.011, d = 0.36, but they did not differ in the level of BR^2^, t(2561) = −1.24, *p* = 0.214, d = 0.18.

For parents who had adopted a child, we identified four confounding variables. They were older, t(2561) = −8.06, *p* < 0.001, d = 0.56, were more likely to live with the adopting co-parent than as a single parent, b = −1.35, *p* < 0.001 or in a step-family, b = −1.16, *p* < 0.001, V = 0.18, had a higher educational level, t(2561) = −6.48, *p* < 0.001, d = 0.45, and had a higher income, t(2561) = −9.81, *p* < 0.001, d = 0.70, than parents without an adopted child. Considering the confounding effects of demographic factors, we successfully matched 130 pairs of parents. We checked the differences between matched and non-matched parents. Matched parents displayed a similar level of PB than non-matched ones, t(2561) = −1.25, *p* = 0.212, d = 0.08, as well as a similar level of BR^2^, t(2561) = 1.61, *p* = 0.107, d = 0.10.

Single parents were significantly different from parents living with the other biological parent or in a step-family in respect of six demographic factors. They were more likely to be a mother, b = 1.02, *p* < 0.001, V = 0.17, were older, t(2561) = −4.95, *p* < 0.001, d = 0.20, had fewer children, t(2561) = 12.22, *p* < 0.001, d = 0.49, had a lower educational level, t(2561) = 4.77, *p* < 0.001, d = 0.19, were less likely to work full-time, b = −0.28, *p* = 0.006, or part-time, b = −0.53, *p* < 0.001, than to have no job, V = 0.10, and had a lower income, t(2285) = −32.90, *p* < 0.001, d = 1.38, than parents living with the other parent or in a step-family. Considering the confounding effects of demographic factors, we successfully matched 109 pairs of parents. We checked the differences between matched and non-matched parents. Matched parents displayed a similar level of PB than non-matched ones, t(2561) = 0.12, *p* = 0.906, d = 0.01, as well as a similar level of BR^2^, t(2561) = 0.52, *p* = 0.601, d = 0.04.

### 3.2. Main Analyses

Descriptive statistics for the level of parental burnout and the balance between risks and resources in the three matched samples, the results of the t-tests, and effect sizes, are presented in Table 1. The level of parental burnout was significantly higher among parents with a child with special needs than in the matched control group of parents, with a medium effect size. The level of balance between risks and resources also tended to be lower for parents with a child with special needs than in the matched control group of parents, with a small effect size. The difference in the mean level of parental burnout was also significant for single parents, with a small effect size. There was no statistical difference between the mean level of parental burnout among parents of an adopted child and their matched control group.

We found a correlation of r = −0.47, *p* < 0.05 (R^2^ = 22%) between the balance between risks and resources and the level of parental burnout among parents with a child with special needs. The correlation was r = −0.69, *p* < 0.001 (R^2^ = 48%) in the matched control group of parents. The difference between the two coefficients was not significant, Z = 1.21, *p* = 0.131. We found a correlation of r = −0.52, *p* < 0.05 (R^2^ = 27%) between the balance between risks and resources and the level of parental burnout among parents who had adopted a child. The correlation was r = −0.49, *p* < 0.001 (R^2^ = 24%) in the matched control group of parents. The difference between the two coefficients was not significant, Z = 0.32, *p* = 0.374. Finally, we found a correlation of r = −0.51, *p* < 0.001 (R^2^ = 26%) between the balance between risks and resources and the level of parental burnout among single parents. The correlation was r = −0.36, *p* < 0.001 (R^2^ = 13%) in the matched control group of parents. The difference between the two coefficients was not significant, Z = −1.44, *p* = 0.075.

Finally, for parental burnout, we found significant mean differences between parents with no vulnerability factor and those who had one, two, or three, F(3;2559) = 14.70, *p* < 0.001. The parents who combined more vulnerability factors had higher mean levels of parental burnout. Pairwise comparisons showed that the difference was significant but small between having no vulnerability factor (M_no vulnerability factor_ = 23.76 SD_no vulnerability factor_ = 25.96) and having one (M_one vulnerability factor_ = 27.68 SD_one vulnerability factor_ = 28.01), t(2430) = −3/56, *p* < 0.000, d = 0.14. It was significant and medium between having no vulnerability factor and having two (M_two vulnerability factors_ = 37.96 SD_two vulnerability factors_ = 36.08), t(1247) = −5.33, *p* < 0.000, d = 0.52. It was significant and large between having no vulnerability factor and having three (M_three vulnerability factors_ = 48.17 SD_three vulnerability factors_ = 36.24), t(1152) = −3.93, *p* < 0.000, d = 0.93. It was significant and medium between having one vulnerability factor and having three, t(1312) = −3.07, *p* = 0.001, d = 0.73. These results suggest a linear relationship between the combination of vulnerability factors and parental burnout.

Similarly, for the balance between risks and resources, we found significant mean differences between parents with no vulnerability factor and those who had one, two, or three, F(3;2559) = 2.96, *p* = 0.031. The parents who combined more vulnerability factors had a lower mean level of the balance between risks and resources. Pairwise comparisons showed that the difference was significant but small between having no vulnerability factor (M_no vulnerability factor_ = 0.35 SD_no vulnerability factor_ = 1.41) and having two (M_two vulnerability factors_ = 0.13, SD_two vulnerability factors_ = 1.14), t(1247) = 1.99, *p* = 0.023, d = 0.20. It was significant and medium between having no vulnerability factor and having three (M_three vulnerability factors_ = −0.24, SD_three vulnerability factors_ = 1.04), t(1252) = 2.18, *p* = 0.015, d = 0.52. It was significant and small (but near to medium) between having one vulnerability factor (M_one vulnerability factor_ = 0.29, SD_one vulnerability factor_ = 1.14) and having three, t(1312) = 1.96, *p* = 0.025, d = 0.47. These results suggested a linear relation between the combination of vulnerability factors and the balance between risks and resources.

## 4. Discussion

The objective of this study was to investigate if having a child with special needs, having adopted a child, or single parenthood, were specific vulnerability factors that significantly increase the risk of parental burnout. We assessed parental burnout and the balance between risks and resources in three specific samples: parents with a child with special needs, parents who had adopted a child, and single parents. We then compared these parents to control groups of parents, taking care to identify confounding variables, strictly match the specific and the control groups on all confounding demographic variables, and estimate the effect of the combined presence of a child with special needs and/or an adopted child and/or single parenthood, on both the balance between risks and resources and parental burnout.

To summarize, the results supported H1 and H2 for parents who had adopted a child. They had similar levels of both parental burnout and the balance between risks and resources as control parents. We did not confirm H1 but we did confirm H2 for single parents, who a had higher level of parental burnout but a similar level of balance between risks and resources as control parents. We did not confirm H1 or H2 for parents with a child with special needs, who displayed a higher level of parental burnout and tended to have a lower level of balance between risks and resources than control parents. Lastly, we confirmed H3, that is, the greater risk of burnout for parents who accumulated specific several vulnerability factors.

Does this finding that adoptive parents do not have higher parental burnout levels than their matched control group, mean that it is not stressful to raise an adopted child? In line with the BR^2^ etiological framework, this finding rather suggests that many adoptive parents in our sample had found resources to compensate for the stress entailed by raising an adopted child and/or that control parents presented other risk factors that made them vulnerable to parental burnout too. The finding also suggests that adopting a child does not in itself trigger a process leading de facto to other risk factors.

By contrast, the finding that parents of children with specific needs have both a more negative balance and higher levels of parental burnout opens the possibility that the mere presence of a child with special needs in the family entails a cascade of other risk factors such as lack of or inadequate social support [6], siblings conflicts and concerns [40,41], couple conflicts [42]; all of these factors are present in BR^2^ because they have themselves proved to be risk factors for parental burnout in other contexts [23,29]. The higher level of parental burnout among these parents may also be explained by the fact that children with special needs may be less easily included in ordinary schools or extracurricular activities, or less autonomous/independent than adopted children, both of which may further increase parents’ anxiety about their children’s future.

The results pertaining to single parents may seem less consistent than those for the two other groups, in that these parents do have a slightly higher level of parental burnout than control parents, but a similar level of balance between risks and resources. Our interpretation of these findings is that they suggest that removing all items related to family composition and co-parenting from the BR^2^ of single parents may not be the best strategy to measure the balance of these parents. We had no alternative in the current study because items pertaining to family composition or co-parenting are phrased in a way that makes it impossible for single parents to respond. However, our results suggest that in future studies, items should be specifically designed for single parents that make it possible to record whether they perceive raising their children alone as a burden or a blessing. It would also be of utmost importance to add items capturing the possible presence of a former co-parent (many single parents are divorced) and the quality of the co-parenting relationship with the latter. These factors are likely to weigh heavily on these parents’ balance.

Beyond the impact of the specific vulnerabilities described above, the results of this study showed that the combination of vulnerability factors is less easy to compensate for and therefore more detrimental in terms of parental burnout. Parents with three vulnerability factors had a lower balance between risks and resources and higher parental burnout than parents with two vulnerabilities, who themselves had a lower balance between risks and resources and higher parental burnout than parents with only one vulnerability factor, and so on. These results, suggesting a linear relationship between the combination of vulnerability factors and parental burnout, fully dovetail with other results obtained by Roskam and Mikolajczak [31] on another sample. The authors examined the cumulative effect of a set of 37 risk factors belonging to five domains (i.e., sociodemographics, particularities of the child, stable traits of the parent, parenting, and family functioning). The presence or absence of each risk was coded as 1 or 0. The sum of the risk factors was then calculated, and related to parental burnout in a community sample of 1540 parents. The results show that the number of risk factors to which a parent is exposed linearly predicts parental burnout regardless of the nature of these factors.

### Limitations and Future Perspectives

A limitation of this study is that we do not have data about the children’s difficulties or diagnoses. As this study was part of a larger one, we had to limit the length of the survey. We did not have the opportunity to go into greater depth for each situation, for example, the type of child’s difficulties for parents having a child with special needs or the time since the child’s diagnosis, and age at diagnosis, the origin, and age of adoption or the specific circumstances that preceded the adoption (e.g., the experience of neglecting the adopted child in the biological family or difficulties with accepting infertility in the adoptive parents) for adopting parents, the type of single parenthood (widowed, single by choice, single after divorce) or the number of years of single parenthood. Now that our results have shown that parents of children with special needs have a higher risk of burnout, further studies are needed to identify which parents of children with special needs are most at risk. This could pave the way to another important avenue for future research which is the investigation of possible mechanisms that are driving the increased risk of burnout among parents having a child with special needs. It should consider the fact that these mechanisms may differ according to the child’s difficulties and diagnosis.

The methodology of this research overcame the limitations of previous studies and provided a more convincing answer to the question of whether there are specific vulnerabilities to parental burnout. Despite this methodological rigor, one could still argue that each specific population has its own risk factors and resources and that BR^2^ as used here for comparison purposes is not a measure capable of accounting for all of these specifics. The results we obtained do not contradict the existence of specific factors, but they support the etiological model according to which any risk can be compensated for by any resource of the same weight (or by several resources of lesser weight), regardless of the nature of these factors.

Parental burnout is based on a common etiological process shared by all parents involved, regardless of their specific situation. Recently validated treatments that have been developed by Brianda and colleagues [43] on the basis of the BR^2^ etiological model, should therefore be equally effective regardless of the parent’s situation. This could be the subject of future investigations by replicating evidence-based interventions in specific populations and groups of parents with different profiles. Furthermore, given the impact of combining several vulnerability factors, it would be interesting for future studies to focus on the treatment of burnout in parents exposed to this combination of vulnerabilities, and ensure that it is effective.

## 5. Conclusions

Based on the current results and comparison of strictly matched samples, we can conclude that parents having a child with special needs displayed higher burnout than the matched control group; parents with a child with special needs displayed a lower balance between parental risk and resources than the marched control group; single parenthood and adoption do not trigger a process leading de facto to other risk factors, but having a child with special needs does; parents who combined specific several vulnerability factors were at greater risk of burnout.

## Figures and Tables

**Table 1 children-10-01131-t001:** Means and Standard Deviations for the Level of Parental Burnout and the Balance Between Risks and Resources in the Three Matched Samples of Parents, Results of Group Comparisons, and Effect Sizes.

	**Parents with a Child with Special Needs** **(*n* = 25)**		**Control Group of Parents** **(*n* = 25)**		** *t* ** **(48)**	** *p* **	**Cohen’s *d***
	** *M* **	** *SD* **	** *M* **	** *SD* **			
PB	47.72	38.10	24.96	27.59	−2.42	0.010	0.68
BR^2^	−0.03	0.90	0.35	0.93	1.48	0.072	0.42
	**Parents with an Adopted Child ** **(*n* = 130)**		**Control Group of Parents ** **(*n* = 130)**		** *t* ** **(258)**	** *p* **	**Cohen’s *d***
	** *M* **	** *SD* **	** *M* **	** *SD* **			
PB	25.63	28.53	26.53	27.65	0.26	0.601	0.03
BR^2^	0.50	1.01	0.33	1.23	−1.22	0.887	0.19
	**Single Parents** **(*n* = 109)**		**Control Group of Parents** **(*n* = 109)**		** *t* ** **(216)**	** *p* **	**Cohen’s *d***
	** *M* **	** *SD* **	** *M* **	** *SD* **			
PB	31.07	27.74	23.03	28.15	−2.13	0.017	0.29
BR^2^	0.19	1.01	0.33	1.09	0.95	0.173	0.13

## Data Availability

Data and syntax are available on OSF https://osf.io/e4xw5/ (accessed on 17 August 2022).

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
