# Peer review of "Parental Burnout in the Context of Special Needs, Adoption, and Single Parenthood"

_children, 2023, doi:10.3390/children10071131_

Round 1
Reviewer 1 Report
A brief summary: Using the Balance between Risks and Resources, BR2) as a theoretical lens, the authors sought to determine if there were specific vulnerability factors that significantly increase the risk of parental burnout. Comparing three categories of parents (i.e. parents with a child with special needs, those with an adopted, and single parents to matched control groups), they found that parents having a child with special needs displayed higher burnout than the matched control group. They also found that parents who combined specific several vulnerability factors were at greater risk of burnout. In my view, these key findings make a significant contribution to our understanding of parental burnout.
General comments: The article is generally well written, with three testable hypotheses. The methods used are relevant for a study of this nature. The authors are commended for using a comprehensive theoretical and clinical framework (the balance between risks and resources) to structure their paper. In their literature review, the authors have adequately consulted relevant sources and have sought to address the existing knowledge gaps on the subject.
Specific comments: (a). I found the first two sentences of the introduction (i.e. lines 27 to 29) irrelevant. The authors should rather consider starting their Introduction with what is currently the third sentence (i.e. sentence stating “It has previously been 29 shown that parents who are chronically stressed”; (b). In line 64, the authors refer to “… the pandemic …”. It is advisable to be specific and refer to this as “… the COVID-19 pandemic …”; (c). In line 192, reference is made to IRB. I suggest that this be written in full; and, (d). The sentence starting in line 409 and ending in line 412 – “The authors examined the cumulative effect a set of risk factors belonging …” ) is not grammatically clear. I suggest the authors attend to this.
The manuscript reads well. I have made some comments in the section above regarding some minor grammatical issues that need attention.
Author Response
Authors’ reply: We would like to express our gratitude to the editor and reviewers for their comments and suggestions, which helped us to strengthen the paper. In order to show how we have addressed each comment/suggestion and to what changes it led in the manuscript, we have broken the reviews down into individual points.
A brief summary: Using the Balance between Risks and Resources, BR2) as a theoretical lens, the authors sought to determine if there were specific vulnerability factors that significantly increase the risk of parental burnout. Comparing three categories of parents (i.e. parents with a child with special needs, those with an adopted, and single parents to matched control groups), they found that parents having a child with special needs displayed higher burnout than the matched control group. They also found that parents who combined specific several vulnerability factors were at greater risk of burnout. In my view, these key findings make a significant contribution to our understanding of parental burnout.
General comments: The article is generally well written, with three testable hypotheses. The methods used are relevant for a study of this nature. The authors are commended for using a comprehensive theoretical and clinical framework (the balance between risks and resources) to structure their paper. In their literature review, the authors have adequately consulted relevant sources and have sought to address the existing knowledge gaps on the subject.
Reply: We thank Reviewer 1 for the positive feedback on our manuscript.
Specific comments: (a). I found the first two sentences of the introduction (i.e. lines 27 to 29) irrelevant. The authors should rather consider starting their Introduction with what is currently the third sentence (i.e. sentence stating “It has previously been 29 shown that parents who are chronically stressed”; (b). In line 64, the authors refer to “… the pandemic …”. It is advisable to be specific and refer to this as “… the COVID-19 pandemic …”; (c). In line 192, reference is made to IRB. I suggest that this be written in full; and, (d). The sentence starting in line 409 and ending in line 412 – “The authors examined the cumulative effect a set of risk factors belonging …” ) is not grammatically clear. I suggest the authors attend to this.
Reply: Thanks for raising our attention on these minor points.
Changes in the manuscript:
- Lines 27-29 have been removed
- Line 64, we refer now to COVID-19 pandemic
- IRB has been written in full, i.e. Institutional Review Board
- The sentence has been rephrased and extended to be clearer: “The authors examined the cumulative effect a set of 37 risk factors belonging to five domains (i. e. sociodemographics, particularities of the child, stable traits of the parent, parenting and family functioning). The presence or absence of each risk was coded as 1 or 0. The sum of the risk factors was then calculated, and related to parental burnout in a community sample of 1,540 parents. The results show that the number of risk factors to which a parent is exposed linearly predicts parental burnout regardless of the nature of these factors.”
Comments on the Quality of English Language
The manuscript reads well. I have made some comments in the section above regarding some minor grammatical issues that need attention.
Reply: We thank Reviewer 1 for the positive feedback on the quality of English language.
Reviewer 2 Report
The manuscript has an appropriate title and structure. The writing style is clear and engaging. The results of this research are relevant and of interest to the scientific community.
The objective of this study is to determine if having a child with special needs, having adopted a child, or single parenthood, were specific vulnerability factors that significantly increase the risk of parental burnout).
The study is based on a clear theoretical framework („The Etiological Framework of Parental Burnout ").
The manuscript includes an informative overview of the methodology. The authors use a methodology that considers the confounding variables and the combination of vulnerabilities in certain families. The methodology of this research overcame the limitations of previous studies and provided a more convincing answer to the question of whether there are specific vulnerabilities to parental burnout in three specific samples: parents with a child with special needs, parents who had adopted a child, and single parents. The authors assessed parental burnout and the balance between risks and resources. They then compared these parents to control groups of parents, taking care to identify confounding variables, strictly match the specific and the control groups on all confounding demographic variables, and estimate the effect of the combined presence of a child with special needs and/or an adopted child and/or single parenthood, on both the balance between risks and resources and parental burnout.
Results are presented in a well-structured manner. The conclusions are consistent with the evidence and arguments presented. All research hypotheses are validated by data and statistical analysis.
Before publication, I suggest that the authors make minor revisions to the paper, considering the following:
The first sentence in the introduction is unclear. Is there a parent's quote missing that the authors refer to claiming that „These are the words of encountered parents of children with special needs, adoptive parents, and single parents respectively "?
Data were collected from a big sample of 2,563 French- and Dutch-speaking parents, but the specific sample of parents who have a child with special needs is quite small (N=168). The description of this specific sample should include a description of the children's difficulties/diagnoses. This is important because different types of children's difficulties can cause different levels of parental stress and parental burnout. The results would be more convincing if the analysis was done on a more precisely defined or more homogeneous sample of parents with children with certain developmental disabilities. In this form, it is necessary to state this both in the discussion of the results and as a limitation of the research.
In the chapter Methodological Approach, it is pointed out that „Studying the true influence of any specific vulnerability factor is challenging because of the cooccurrence of multiple risk factors which are usually interrelated. For example, having a child with special needs is often related to financial difficulties, child adoption is more frequent among higher-educated parents, and single parenthood is often associated with lower household income. "
Along with the mentioned examples, research that confirms this should be cited. In addition, it is confusing why a higher level of education is mentioned as an additional risk for the greater vulnerability of the parents when parenting an adopted child. It would be more logical to see this as an other resource, and as an additional risk to list, for example, specific circumstances that preceded the adoption (the experience of neglecting the adopted child in the biological family or difficulties with accepting infertility in the adoptive parents).
Also, it should be kept in mind that adoption very often involves parenting children with developmental disabilities. The description of the sample does not indicate how many adoptive parents had children with developmental disabilities. It was also not stated whether and in what way the possible overlap of those sample categories was considered during data processing.
The authors use extensive self-citation of their work - out of a total of 43 sources in the list of references, 10 are self-citations (by Roskam and/or Mikolajczak)
In reference number 20 in the list of references, the number of pages is not indicated correctly (Pinderhughes, E.E.D.M. Brodzinsky. Parenting in Adoptive Families. In: Handbook of Parenting. 3rd ed; MH Bornstein 508 (eds); Routledge: London, UK, 2019; 322-67.)
Author Response
Authors’ reply: We would like to express our gratitude to the editor and reviewers for their comments and suggestions, which helped us to strengthen the paper. In order to show how we have addressed each comment/suggestion and to what changes it led in the manuscript, we have broken the reviews down into individual points.
The manuscript has an appropriate title and structure. The writing style is clear and engaging. The results of this research are relevant and of interest to the scientific community.
The objective of this study is to determine if having a child with special needs, having adopted a child, or single parenthood, were specific vulnerability factors that significantly increase the risk of parental burnout).
The study is based on a clear theoretical framework („The Etiological Framework of Parental Burnout ").
The manuscript includes an informative overview of the methodology. The authors use a methodology that considers the confounding variables and the combination of vulnerabilities in certain families. The methodology of this research overcame the limitations of previous studies and provided a more convincing answer to the question of whether there are specific vulnerabilities to parental burnout in three specific samples: parents with a child with special needs, parents who had adopted a child, and single parents. The authors assessed parental burnout and the balance between risks and resources. They then compared these parents to control groups of parents, taking care to identify confounding variables, strictly match the specific and the control groups on all confounding demographic variables, and estimate the effect of the combined presence of a child with special needs and/or an adopted child and/or single parenthood, on both the balance between risks and resources and parental burnout.
Results are presented in a well-structured manner. The conclusions are consistent with the evidence and arguments presented. All research hypotheses are validated by data and statistical analysis.
Reply: We thank Reviewer 2 for the positive feedback on our manuscript.
Before publication, I suggest that the authors make minor revisions to the paper, considering the following:
Reply: Thanks for raising our attention on these minor points.
The first sentence in the introduction is unclear. Is there a parent's quote missing that the authors refer to claiming that „These are the words of encountered parents of children with special needs, adoptive parents, and single parents respectively "?
Change in the manuscript: Lines 27-29 have been removed.
Data were collected from a big sample of 2,563 French- and Dutch-speaking parents, but the specific sample of parents who have a child with special needs is quite small (N=168). The description of this specific sample should include a description of the children's difficulties/diagnoses. This is important because different types of children's difficulties can cause different levels of parental stress and parental burnout. The results would be more convincing if the analysis was done on a more precisely defined or more homogeneous sample of parents with children with certain developmental disabilities. In this form, it is necessary to state this both in the discussion of the results and as a limitation of the research.
Reply: Unfortunately, we do not have data about the children’s difficulties or diagnoses. As this study was part of a larger one, we had to limit the length of the questionnaire. We did not have the opportunity to go into greater depth for each situation, for example, the type of child's difficulties for parents having a child with special needs, the origin and age of adoption for adopting parents, the type of single parenthood (widowed, single by choice, single after divorce) or the number of years of single parenthood. This is a limitation of the study that we have now added to the discussion.
Change in the manuscript: “A limitation of this study is that we do not have data about the children’s difficulties or diagnoses. As this study was part of a larger one, we had to limit the length of the survey. We did not have the opportunity to go into greater depth for each situation, for example, the type of child's difficulties for parents having a child with special needs or the time since the child’s diagnosis, and age at diagnosis, the origin and age of adoption or the specific circumstances that preceded the adoption (e.g. the experience of neglecting the adopted child in the biological family or difficulties with accepting infertility in the adoptive parents) for adopting parents, the type of single parenthood (widowed, single by choice, single after divorce) or the number of years of single parenthood. Now that our results have shown that parents of children with special needs have a higher risk of burnout, further studies are needed to identify which parents of children with special needs are most at risk.”
In the chapter Methodological Approach, it is pointed out that „Studying the true influence of any specific vulnerability factor is challenging because of the cooccurrence of multiple risk factors which are usually interrelated. For example, having a child with special needs is often related to financial difficulties, child adoption is more frequent among higher-educated parents, and single parenthood is often associated with lower household income. " Along with the mentioned examples, research that confirms this should be cited. In addition, it is confusing why a higher level of education is mentioned as an additional risk for the greater vulnerability of the parents when parenting an adopted child. It would be more logical to see this as an other resource, and as an additional risk to list, for example, specific circumstances that preceded the adoption (the experience of neglecting the adopted child in the biological family or difficulties with accepting infertility in the adoptive parents).
Reply: Thanks for getting our attention to this important point. We realized that the wording of this paragraph was not optimal. Change in the manuscript: “Studying the true influence of any specific vulnerability factor is challenging because of the co-occurrence of multiple risk factors which are can be interrelated. For example, having a child with special needs may be related to financial difficulties, and single parenthood may be associated with lower household income.”
Reply: Moreover, we included the specific circumstances preceding adoption, in the limitations section. Change in the manuscript: “We did not have the opportunity to go into greater depth for each situation, for example, (…), the origin and age of adoption or the specific circumstances that preceded the adoption (e.g. the experience of neglecting the adopted child in the biological family or difficulties with accepting infertility in the adoptive parents) for adopting parents,…”
Also, it should be kept in mind that adoption very often involves parenting children with developmental disabilities. The description of the sample does not indicate how many adoptive parents had children with developmental disabilities. It was also not stated whether and in what way the possible overlap of those sample categories was considered during data processing.
Reply: Thanks for this relevant comment. The information is now given in the description of the sample. Change in the manuscript: “As regards the presence of two of the three vulnerability factors under study, 1.7% were single parents having an adopted child, 3.6% were single parents having a child with special needs, and 0.7% were parents having an adopted child and a child with special needs.”
The authors use extensive self-citation of their work - out of a total of 43 sources in the list of references, 10 are self-citations (by Roskam and/or Mikolajczak)
Reply: This is due to the fact that we are leaders in this recent field of research (i.e. Parental Burnout) and have conducted a large number of studies in comparison with other research teams. We created the International Investigation of Parental Burnout (IIPB), a consortium of over 40 countries, to stimulate research at other universities around the world.
In reference number 20 in the list of references, the number of pages is not indicated correctly (Pinderhughes, E.E.D.M. Brodzinsky. Parenting in Adoptive Families. In: Handbook of Parenting. 3rd ed; MH Bornstein 508 (eds); Routledge: London, UK, 2019; 322-67.)
Reply: We corrected the reference.
Reviewer 3 Report
This paper investigates a few characteristics that are considered potential risk factors of parental burnout. Such a study is important, not only because it is of interest to identify parents at risk, but it is also important to consider if additional resources might reduce the risk of burnout in exposed parents. Nevertheless, there are a few corrections and clarifications that need to be made before the manuscript is ready for publication. Further, I also have a few suggested additional analyses.
Major Comments:
How are special needs defined in this study? I think it is autism and mental retardation, serious illnesses such as Type 1 diabetes mellitus, or recovering from a brain tumor but this is not stated clearly enough in the paper.
As far as I understand the parent’s characteristics are assessed at the time of the survey and not before the child’s diagnosis, which means that some of them could be a consequence of living with a child with special needs. I am mainly thinking of working full time, and income. If a characteristic is a consequence of the exposure, then it is not a confounder and matching on it might underestimate the true difference between the groups. To make sure this is not the case you need to run a sensitivity analysis without matching on variables that could be on the casual pathway between the exposure and the outcome. Possibly this will also increase your sample size which is very low and perhaps not representative of the whole group for the “child with special needs” - analysis.
The exposed who were not matched need to be described so that the reader can evaluate if this group was any different than the exposed who were matched. I am primarily thinking of whether the level of burn out was higher, lower or similar in the matched and unmatched. This is important in order to understand the generalizability of the results.
The results section says: “…had a lower educational level, t(1465) = 2.85, p = .004, d = .21, were less likely to work full-time than to have no job, b = -0.49, p = .007, V = .08, and had a higher income”. I am surprised that the income was higher when the level of education was lower and they were less likely to work full time, please double-check that this is correct, and if correct please check if this is true for the whole data set which makes me wonder about the generalizability of the results from this study.
If available, please also report the time since the child’s diagnosis, and/or age at diagnosis
In the discussion, the authors claim that they “confirmed H1 and H2” but a non-significant result is not the same as a confirmation of the null hypothesis, or in other words, absences of evidence is not evidence of absence. Please phrase is a correct way.
A discussion of possible unmeasured confounders is missing. It could be that your genetic risk of burn out is also associated with having a child with some (or all?) of the special needs that are investigated. See for example https://doi.org/10.2337/dc21-1347 where the author suggested a common familial risk between childhood-onset type 1 diabetes and depression, anxiety, and stress-related disorders. I am not suggestion that this is the cause of the found difference, simply that it could be slightly biases.
The possible mechanism could be further discussed and explored in subgroup analysis. I am thinking about one suggested mechanism “…with special needs may be less easily included in ordinary schools or extracurricular activities”. This might be true for children with neuropsychiatric disabilities, while children with type 1 diabetes might very well be like any other child in school or at activities. On the other hand, type 1 diabetes requires constant monitoring in another way than other diagnoses. If, and only if, there are enough subjects in each group it would be interesting to see if the effects size is similar in the subgroups. In the case of larger differences, it could be a piece of the puzzle to understand the mechanisms that are driving the increased risk of burnout.
Minor comments:
Could two parents in the sample be parents to the same child?
It says the age range is 10-73, and then later: “Parents had to be at least 18 years old”. Please correct.
Avoid the term “determine” it is too strong in this context, and these kinds of associations might not be constant over time but may depend on the resources proved by the society to expose children and families (as also suggested by the author).
The link to data and scripts was not working so I could not review that part.
Author Response
Authors’ reply: We would like to express our gratitude to the editor and reviewers for their comments and suggestions, which helped us to strengthen the paper. In order to show how we have addressed each comment/suggestion and to what changes it led in the manuscript, we have broken the reviews down into individual points.
This paper investigates a few characteristics that are considered potential risk factors of parental burnout. Such a study is important, not only because it is of interest to identify parents at risk, but it is also important to consider if additional resources might reduce the risk of burnout in exposed parents. Nevertheless, there are a few corrections and clarifications that need to be made before the manuscript is ready for publication. Further, I also have a few suggested additional analyses.
Reply: We thank Reviewer 1 for the positive feedback and interesting comments on our manuscript.
Major Comments:
How are special needs defined in this study? I think it is autism and mental retardation, serious illnesses such as Type 1 diabetes mellitus, or recovering from a brain tumor but this is not stated clearly enough in the paper.
Reply: Unfortunately, we do not have data about the children’s difficulties or diagnoses. As this study was part of a larger one, we had to limit the length of the questionnaire. We did not have the opportunity to go into greater depth for each situation, for example, the type of child's difficulties for parents having a child with special needs, the origin and age of adoption for adopting parents, the type of single parenthood (widowed, single by choice, single after divorce) or the number of years of single parenthood. This is a limitation of the study that we have now added to the discussion.
Change in the manuscript: “A limitation of this study is that we do not have data about the children’s difficulties or diagnoses. As this study was part of a larger one, we had to limit the length of the survey. We did not have the opportunity to go into greater depth for each situation, for example, the type of child's difficulties for parents having a child with special needs or the time since the child’s diagnosis, and age at diagnosis, the origin and age of adoption or the specific circumstances that preceded the adoption (e.g. the experience of neglecting the adopted child in the biological family or difficulties with accepting infertility in the adoptive parents) for adopting parents, the type of single parenthood (widowed, single by choice, single after divorce) or the number of years of single parenthood. Now that our results have shown that parents of children with special needs have a higher risk of burnout, further studies are needed to identify which parents of children with special needs are most at risk.”
As far as I understand the parent’s characteristics are assessed at the time of the survey and not before the child’s diagnosis, which means that some of them could be a consequence of living with a child with special needs. I am mainly thinking of working full time, and income. If a characteristic is a consequence of the exposure, then it is not a confounder and matching on it might underestimate the true difference between the groups. To make sure this is not the case you need to run a sensitivity analysis without matching on variables that could be on the casual pathway between the exposure and the outcome. Possibly this will also increase your sample size which is very low and perhaps not representative of the whole group for the “child with special needs” - analysis.
Reply: Thanks for this interesting suggestion. We did sensitivity analyses for the group of parents with a child with special needs, with and without working Status, and with and without income in the matching. The sample sizes are slightly larger, i.e. 51 and 58 respectively. But as might be expected, the results change only slightly, as the effect sizes reported for comparisons involving confounding variables are small. We think that mentioning these additional analyses in the article would add a great deal of complexity without any significant benefit.
The exposed who were not matched need to be described so that the reader can evaluate if this group was any different than the exposed who were matched. I am primarily thinking of whether the level of burn out was higher, lower or similar in the matched and unmatched. This is important in order to understand the generalizability of the results.
Reply: We added the information about the differences between matched and non-matched participants, for each of the three subsamples (i.e., special needs, adoption, single parenthood), in the Preliminary Analysis section.
Change in the manuscript: “We checked the differences between matched and non-matched parents. Matched parents displayed a higher level of PB than non-matched ones, t(2561) = 2.55, p = .011, d = .36, but they did not differ in the level of BR2, t(2561) = -1.24, p = .214, d = .18. (…) We checked the differences between matched and non-matched parents. Matched parents displayed a similar level of PB than non-matched ones, t(2561) = -1.25, p = .212, d = .08, as well as a similar level of BR2, t(2561) = 1.61, p = .107, d = .10. (…) We checked the differences between matched and non-matched parents. Matched parents displayed a similar level of PB than non-matched ones, t(2561) = .12, p = .906, d = .01, as well as a similar level of BR2, t(2561) = 0.52, p = .601, d = .04.”
The results section says: “…had a lower educational level, t(1465) = 2.85, p = .004, d = .21, were less likely to work full-time than to have no job, b = -0.49, p = .007, V = .08, and had a higher income”. I am surprised that the income was higher when the level of education was lower and they were less likely to work full time, please double-check that this is correct, and if correct please check if this is true for the whole data set which makes me wonder about the generalizability of the results from this study.
Reply: We carefully checked the results and confirm that they are correct. We asked the participants to report their net monthly incomes. In Belgium where the study has been conducted, parents with a child with special needs benefit from fairly high allowances and also tax reductions, which may easily explain why they have slightly higher incomes on average than parents in a similar situation in terms of work pattern and level of education, but without children with special needs.
If available, please also report the time since the child’s diagnosis, and/or age at diagnosis
Reply: As stated earlier, we do not have data about the children’s difficulties or diagnoses. As this study was part of a larger one, we had to limit the length of the questionnaire. We did not have the opportunity to go into greater depth for each situation, for example, the type of child's difficulties for parents having a child with special needs, the origin and age of adoption for adopting parents, the type of single parenthood (widowed, single by choice, single after divorce) or the number of years of single parenthood. This is a limitation of the study that we have now added to the discussion.
Change in the manuscript: “A limitation of this study is that we do not have data about the children’s difficulties or diagnoses. As this study was part of a larger one, we had to limit the length of the survey. We did not have the opportunity to go into greater depth for each situation, for example, the type of child's difficulties for parents having a child with special needs or the time since the child’s diagnosis, and age at diagnosis, the origin and age of adoption or the specific circumstances that preceded the adoption (e.g. the experience of neglecting the adopted child in the biological family or difficulties with accepting infertility in the adoptive parents) for adopting parents, the type of single parenthood (widowed, single by choice, single after divorce) or the number of years of single parenthood. Now that our results have shown that parents of children with special needs have a higher risk of burnout, further studies are needed to identify which parents of children with special needs are most at risk.”
In the discussion, the authors claim that they “confirmed H1 and H2” but a non-significant result is not the same as a confirmation of the null hypothesis, or in other words, absences of evidence is not evidence of absence. Please phrase is a correct way.
Reply: We rephrased the sentence as follows: “To summarize, the results supported H1 and H2 for parents who had adopted a child.”
A discussion of possible unmeasured confounders is missing. It could be that your genetic risk of burn out is also associated with having a child with some (or all?) of the special needs that are investigated. See for example https://doi.org/10.2337/dc21-1347 where the author suggested a common familial risk between childhood-onset type 1 diabetes and depression, anxiety, and stress-related disorders. I am not suggestion that this is the cause of the found difference, simply that it could be slightly biases.
Reply: Thank you for this fascinating suggestion. There are potentially many other unmeasured confounders that have not been considered. This one is very particular and relates to a specific type of diagnosis. And the confounding variables could actually be different for different diagnoses. Given the lack of precision regarding these diagnoses in our sample (see above), we feel that introducing such an element into the discussion would be of little relevance, and would go far beyond the aims of this study.
The possible mechanism could be further discussed and explored in subgroup analysis. I am thinking about one suggested mechanism “…with special needs may be less easily included in ordinary schools or extracurricular activities”. This might be true for children with neuropsychiatric disabilities, while children with type 1 diabetes might very well be like any other child in school or at activities. On the other hand, type 1 diabetes requires constant monitoring in another way than other diagnoses. If, and only if, there are enough subjects in each group it would be interesting to see if the effects size is similar in the subgroups. In the case of larger differences, it could be a piece of the puzzle to understand the mechanisms that are driving the increased risk of burnout.
Reply: This is an essential avenue for future research. However, we do not have the data to create subgroups. Nor do we have any solid hypotheses to predict the underlying mechanisms. This would require another literature review, a data collection in which the mediating or moderating processes would be properly assessed, and analyses going beyond group comparisons. We have added this in the future perspectives at the end of the article.
Change in the manuscript: “(…) Further studies are needed to identify which parents of children with special needs are most at risk. This could pave the way to another important avenue for future research that is the investigation of possible mechanisms that are driving the increased risk of burnout among parents having a child with special needs. And it should consider the fact that these mechanisms may differ according to the child's difficulties and diagnosis.”
Minor comments:
Reply: Thanks for raising our attention on these minor points.
Could two parents in the sample be parents to the same child?
Reply: The completion of the questionnaire being anonym, we cannot rule out this possibility.
It says the age range is 10-73, and then later: “Parents had to be at least 18 years old”. Please correct.
Reply: We apologize for the error. The age range is 19-73.
Avoid the term “determine” it is too strong in this context, and these kinds of associations might not be constant over time but may depend on the resources proved by the society to expose children and families (as also suggested by the author).
Reply: The three occurrences of “determine” have been replaced by “investigate”.
The link to data and scripts was not working so I could not review that part.
Reply: The OSF link is now public. And we updated the syntax according to the revision of the article.
Round 2
Reviewer 3 Report
The authors have addressed all of my comments or provided explanations when this was not possible.